# Functional Analysis of a Novel, Non-Canonical *RPGR* Splice Variant Causing X-Linked Retinitis Pigmentosa

**DOI:** 10.3390/genes14040934

**Published:** 2023-04-18

**Authors:** Samuel Koller, Tim Beltraminelli, Jordi Maggi, Agnès Wlodarczyk, Silke Feil, Luzy Baehr, Christina Gerth-Kahlert, Moreno Menghini, Wolfgang Berger

**Affiliations:** 1Institute of Medical Molecular Genetics, University of Zurich, 8952 Schlieren, Switzerlandberger@medmolgen.uzh.ch (W.B.); 2Department of Ophthalmology, Institute of Clinical Neurosciences of Southern Switzerland, Ente Ospedaliero Cantonale (EOC), 6962 Lugano, Switzerland; 3Department of Ophthalmology, University Hospital, University of Zurich, 8091 Zurich, Switzerland; 4Zurich Center for Integrative Human Physiology (ZIHP), University of Zurich, 8057 Zurich, Switzerland; 5Neuroscience Center Zurich (ZNZ), University and ETH Zurich, 8057 Zurich, Switzerland

**Keywords:** RPGR, retinitis pigmentosa, non-canonical splice variant, minigene

## Abstract

X-linked retinitis pigmentosa (XLRP) caused by mutations in the *RPGR* gene is one of the most severe forms of RP due to its early onset and intractable progression. Most cases have been associated with genetic variants within the purine-rich exon ORF15 region of this gene. *RPGR* retinal gene therapy is currently being investigated in several clinical trials. Therefore, it is crucial to report and functionally characterize (all novel) potentially pathogenic DNA sequence variants. Whole-exome sequencing (WES) was performed for the index patient. The splicing effects of a non-canonical splice variant were tested on cDNA from whole blood and a minigene assay. WES revealed a rare, non-canonical splice site variant predicted to disrupt the wildtype splice acceptor and create a novel acceptor site 8 nucleotides upstream of RPGR exon 12. Reverse-transcription PCR analyses confirmed the disruption of the correct splicing pattern, leading to the insertion of eight additional nucleotides in the variant transcript. Transcript analyses with minigene assays and cDNA from peripheral blood are useful tools for the characterization of splicing defects due to variants in the RPGR and may increase the diagnostic yield in RP. The functional analysis of non-canonical splice variants is required to classify those variants as pathogenic according to the ACMG’s criteria.

## 1. Introduction

Retinitis pigmentosa (RP) is the most common form of inherited retinal dystrophy (IRD) disease, affecting 1 in about 3500–4000 individuals [1,2]. RP has been associated with variants in an exceptionally large list of genes (RetNet^TM^, https://sph.uth.edu/retnet/, accessed on 17 March 2023). Autosomal inherited RP has been described to be caused by recessive variants in at least 44 genes or by dominant variants in 23 genes (RetNet^TM^). In a large IRD cohort, it was found that autosomal dominant inheritance represented around 20% of molecularly diagnosed families, while autosomal recessive inheritance was found in about 64% of the molecularly diagnosed families [3].

X-linked retinitis pigmentosa (XLRP) is one of the most severe forms of retinitis pigmentosa (RP), frequently characterized by strongly reduced retinal function in most affected males before 10 years of age and progression to legal blindness by the third or fourth decade of life [4,5,6,7,8]. XLRP is an orphan disease with a worldwide prevalence of approximately 1:40,000 [6]. It is estimated that XLRP accounts for 7–15% of all RP cases [1]. Furthermore, as recognized several decades ago, female carriers are sometimes symptomatic [9]. The carrier state is typically highly variable and ranges from a normal fundus appearance, or the presence of only the so-called tapetal-like reflex to widespread pigmentary retinopathy resembling a male-pattern disease severity [10].

Three different genes have been linked to XLRP, with the pigmentosa GTPase regulator (*RPGR*) being responsible for more than 60–80% of the cases [11]. The remaining 20–40% of cases are caused by variants in *RP2* [12] or *OFD1* [13].

The *RPGR* was discovered in a patient with XLRP, retinitis pigmentosa-3 (RP3) and could, in a first attempt, be assigned to a genomic segment of less than 1000 kb on the short arm of the human X chromosome (Xp21.1-p11.4, [14]). Subsequently, the *RPGR* cDNA was characterized as a predicted gene product with homology to the regulator of chromosome condensation 1 (RCC1), the guanine nucleotide exchange factor (GEF) of the Ras-like GTPase Ran [15]. The transcription pattern is highly complex, as this gene has been found to undergo extensive differential splicing, generating a number of transcript isoforms that vary according to the tissue of origin [16,17].

The *RPGR* is ubiquitously expressed in ciliated cells throughout the body. The canonical isoform is composed of 19 exons, which is termed the constitutive variant (RPGR^Ex1–19^). It codes for a protein of 815 amino acid residues that localize at the base (the transition zone) of the cilia (primary and motile) or at the centrioles in dividing cells [18]. In the retina, RPGR^Ex1–19^ was found in the connecting cilium of photoreceptor cells, whose function is crucial in protein transport across the outer segment [18].

The retina-specific RPGR isoform, RPGR^ORF15^, encodes a protein consisting of 1152 amino acids. Similar to the constitutive variant, RPGR^ORF15^ also localizes in the photoreceptor-connecting cilium [19]. Moreover, some studies suggest that RPGR^ORF15^ is found in the outer segment of photoreceptor cells in humans but not in mice [20,21]. This isoform contains a highly repetitive Glu-Gly-rich exon (designated exon ORF15), which represents a mutational hotspot [13]. Notably, among the more than 300 known sequence variants in the *RPGR* currently reported in the Human Gene Mutation Database, more than 50% are located within exon ORF15 and lead to frameshift or premature termination codons [5,22]. Similarly, ClinVar currently contains 510 pathogenic and 231 likely pathogenic variants in the *RPGR*; of these, 369 (72%) and 125 (54%) localize to exon ORF15, respectively. Interestingly, only four pathogenic and one likely pathogenic variants have been reported in exons exclusively included in the constitutive form (exons 16–19).

Despite the large number of pathogenic variants identified in exon ORF15, the analysis of this region is challenging due to its highly repetitive nature. We have recently shown that conventional bioinformatic pipelines can result in false negatives and positives and that alternative analysis pipelines can help improve sensitivity and specificity [23,24].

Recent years have seen unprecedented developments in the treatment landscape for RP, with more than ten clinical trials (source: www.clinicaltrials.gov, access on 12 March 2023) currently testing a gene therapy strategy for XLRP. It is, therefore, of high importance to report and characterize all novel potentially pathogenic *RPGR* variants to contribute to the continuously increasing body of scientific knowledge and ultimately help in the quest of developing a therapy for RP.

Here, we describe a novel *RPGR* variant in intron 11 (NM_001034853.1:c.1415-9A>G), leading to a frameshift and a premature stop codon in its transcripts, identified in a 12-year-old male patient suffering from X-linked RP.

## 2. Materials and Methods

### 2.1. Clinical Examinations

The index patient and his affected mother underwent a comprehensive eye examination at the Department of Ophthalmology, the University Hospital of Zurich. Phenotype quantification was performed using spectral-domain optical coherence tomography (Spectralis, Heidelberg Engineering GmbH, Heidelberg, Germany), ultra-widefield retinal imaging (Optomap, Optos plc, Dunfermline, UK), kinetic visual field testing (I:4e and V:4e isopters, Octopus 900 device, Haag Streit, Köniz, Switzerland), and full-field electroretinogram (ERG) recording using DTL electrodes on an Espion system (Diagnosys LLC, Lowell, MA, USA), according to the standards of the International Society for Clinical Electrophysiology of Vision (ISCEV) [25].

### 2.2. Genetic Testing

DNA from blood samples of the index patient and the mother was extracted using a Chemagic DNA Blood kit (Perkin Elmer, Waltham, MA, USA). Whole-exome sequencing (WES) was performed for the index patient using IDT’s Exome kit v2 and additional IDT *RPGR* spike-in probes (Discovery Pool RPGR_15, 54 oligos), according to the manufacturer’s instructions.

Reads were aligned to the human reference genome (hg19), and variant calling was performed according to a Genome Analysis Toolkit (GATK). Best Practices Workflows for germline short variant discovery for single-sample data were used [26]. Briefly, raw unmapped reads were mapped to the hg19 reference sequence using a Burrows–Wheeler Aligner (BWA) v0.7.17 (https://bio-bwa.sourceforge.net/, accessed on 17 March 2023). Subsequently, duplicate reads were tagged using a Picard v2.27.2 MarkDuplicates tool, and the base quality scores were recalibrated using GATK v4.2.6.1 BaseRecalibrator and ApplyBQSR. The alignment file (BAM) was validated using Picard ValidateSamFiles.

Subsequently, variant calling was carried out using GATK HaplotypeCaller, and the identified variants were assigned a score by running GATK CNNScoreVariants in the 2D model setting and GATK FilterVariantTranches. Finally, variant annotation was performed with Alamut Batch v1.11 (Sophia Genetics, Saint Sulpice, Switzerland) for genes (*n* = 241) previously described as being involved in retinal disorders [27].

Filtering and prioritization of variants were performed according to the following strategy. Only variants with a frequency of less than 1% in gnomAD (total allele frequency) and one or more of the following criteria: CADD PHRED [28] with a score higher than 20, a distance to the nearest splice site of within 10 nucleotides, a local splice effect, as predicted by Alamut Batch v1.11, or published in the HGMD (HGMD Professional 2022.4) as a DM (disease-causing mutation), were considered.

Candidate variants were confirmed by Sanger sequencing. Additionally, segregation analysis for candidate variants was performed by Sanger sequencing, as previously described [29].

### 2.3. Functional Analyses

The effect of a candidate variant in intron 11 of the *RPGR* on splicing was first functionally tested in a cellular system using a minigene construct and secondly characterized in RNA extracted from the peripheral blood of the index patient and his mother.

The minigene construct used in this study contains the human *RHO* sequence spanning exons 3 to 5 and has been described in detail previously [29,30]. In brief: the human genomic region encompassing exons 3 to 5 of the gene *RHO* was cloned into the pcDNA3.1 backbone (Invitrogen). An artificial start codon was introduced in the sequence of *RHO* exon 3. Exon 4 of *RHO*, as well as part of the flanking introns, were excised by restriction enzyme digestion with *Pfl*MI and *Eco*NI from the construct and replaced by the human genomic region encompassing exon 12 and part of the flanking introns (introns 11 and 12, NM_001034853.1:c.1415-397_1507-124) of the gene, *RPGR*.

Minigene constructs, with the reference sequence or with the candidate variant, were transfected into HEK293T cells. RNA was extracted and analyzed, as previously described [29,31]. Primers specific to *RHO* exons 3 and 5 were used as forward and reverse primers for the characterization of splicing patterns, respectively (Appendix A).

In addition, RNA was extracted from blood collected in PAXgene Blood RNA tubes (PreAnalytiX, Hombrechtikon, Switzerland) and reverse-transcribed into cDNA, as previously reported [29]. Consequently, primers specific to *RPGR* exons 11 and 13 were used to amplify the region of interest from cDNA with a HOT FIREPol kit (Solis Biodyne, Tartu, Estonia), according to the manufacturer’s instructions (Appendix A). Finally, amplified fragments were separated by agarose gel (1%) electrophoresis and characterized by Sanger sequencing, as previously described [29].

## 3. Results

### 3.1. Clinical Presentation

The index patient is a 12-year-old healthy boy from Portugal with nyctalopia but subjectively preserved peripheral visual fields. Visual acuity was 0.4 and 0.5 decimal Snellen in the right and left eye, respectively. He is myopic, with a spherical equivalent (SEQ) of −6.875 in the right eye and −5.875 diopters in the left eye. The anterior segment examination was normal, while the dilated ophthalmoscopy revealed diffuse pigmentary retinopathy with bone spicule-like pigmentation, mild vascular attenuation, and a slightly pale optic disc in both eyes (Figure 1A). Fundus autofluorescence (FAF) showed corresponding reduced autofluorescence with a ring-shaped area of increased autofluorescence centered on the fovea (Figure 1B), corresponding to the loss of the photoreceptor complex visible in the macular OCT scan, without evidence of macular edema (Figure 1C). Kinetic visual fields (Figure 1D) demonstrated a reduced sensitivity with severe constriction to smaller isopters. The ERG revealed severely reduced scotopic, more than photopic responses (Figure 1E). The last follow-up at the age of 14 years did not show a deterioration of visual function.

Examination of the patient’s mother at the age of 47 years showed a reduced distance visual acuity of hand movements due to a dense cataract in the right eye and 0.6 decimal Snellen equivalent in the left eye. Widefield fundus photography and autofluorescence demonstrated a tapetal-like reflex, as illustrated in Figure 1F–H. An ERG, performed at the age of 50 years, revealed an asymmetric rod–cone dysfunction, which may be partially related to the asymmetric lens opacity. All the findings were compatible with the status of a female carrier of an X-linked RP-causing genetic variant [32].

### 3.2. Genetic Testing of the Patient and His Mother

Filtering of the WES data was performed according to the criteria described in the Methods section. The filtering strategy resulted in six variants in six different genes. All the variants were in a heterozygous state except for the one in the *RPGR*, which was in a hemizygous state in the affected patient (Table 1) and heterozygous in his mother.

The variants in *CC2D2A* and *INPP5E* were retained during filtering due to their positions close to the nearby splice sites. However, both of them showed no effect on splicing, as predicted by different splice prediction algorithms (SpliceAI or built-in AlamutVisualPlus v1.2.1 (Sophia Genetics) tools [33]; SpliceSiteFinder-like, MaxEntScan [34], NNSPLICE [35], and GeneSplicer [36]). The variant in *GRM6* was predicted to create a new acceptor site by Alamut Batch splicing prediction tools. However, *GRM6* variants are inherited in an autosomal recessive manner, and they have been described to lead to a different clinical phenotype (congenital stationary night blindness) [37]. Similarly, the variant in *TMEM67* was within the filtering criteria due to its CADD score. However, variants in this gene have been described to cause syndromic ciliopathies, such as Joubert syndrome [38]. Finally, the variant in *NRL* was retained due to its CADD score. Variants in *NRL* have been described in autosomal dominant retinitis pigmentosa patients [39] and may, therefore, have the potential to explain the patient’s phenotype, as the frequency is quite rare in gnomAD (1/250,000 alleles). However, the predictions are contradictory, ranging from benign to pathogenic. For these reasons, these variants were excluded as the disease-causing variant in this family.

The hemizygous variant in the *RPGR* was predicted to act as a non-canonical splice site (NCSS) variant upstream of exon 12 (NM_001034853.1:c.1415-9A>G, Appendix A). [40] This NCSS variant was not found in the gnomAD database and is predicted to create a novel splice acceptor site 8 nucleotides upstream of exon 12, as well as weaken the existing wildtype acceptor by all the prediction tools on AlamutVisualPlus v1.2.1 (Sophia Genetics) (SpliceSite Finder-like, MaxEntScan, NNSPLICE, and GeneSplicer) (Figure 2A). Similarly, SpliceAI [41] predicted acceptor gain (1 bp downstream from the variant) and acceptor loss (9 bp downstream from the variant), with scores of 0.98 and 0.95, respectively. However, the ACMG (American College of Medical Genetics and Genomics) classified this variant as of unknown significance (VUS, ACMG class 3) due to its localization outside of the canonical splice dinucleotide region.

The clinical findings in the mother of the index patient further support an X-linked inheritance. The presence of this variant in the index and his mother was confirmed by Sanger sequencing (Appendix A, only the mother is shown). No potentially pathogenic variant was found in addition in the patients’ ORF15 region, which was well covered in the WES data by an average of 160 reads, with a minimum coverage of 49 reads (Appendix A). For these reasons, this *RPGR* NCSS variant was identified as the most likely disease-causative variant in this family.

### 3.3. Functional Characterization of an Intronic Sequence Variant

Sequencing of the cDNA derived from HEK293T cells transfected with the variant-containing minigene construct (*RHO* exon 3–introns flanked *RPGR* exon 12–*RHO* exon 5) revealed that the non-canonical splice variant extended exon 12 of the *RPGR* mRNA by 8 nucleotides at the 5′-end (Figure 2B,C). The insertion is out-of-frame and results in a premature stop codon (p.Asp472Valfs*7). Interestingly, Sanger sequencing revealed partial *RPGR* exon 12-skipping, as can be seen by overlapping sequences in both the variant and reference constructs. This was also detected by gel electrophoresis, as shown by a smaller fragment of 92 bp in size and also in lower amounts (Figure 2B and Appendix A). It is important to note that the correctly spliced transcript (without an additional 8 nt) could not be detected by Sanger sequencing in the spliced transcripts of the variant minigene construct. Thus, the variant results in aberrant splicing in the vast majority of the transcripts.

We were able to amplify RPGR exon 11 to 13 in the cDNA derived from a peripheral blood sample from the index patient and his mother (Appendix A), confirming that the RPGR is expressed at low levels in whole blood, as reported in the GTEx database [42]. Similarly, subsequent sequencing of this PCR product confirmed the findings of the minigene assay with the frameshifting insertion of 8 nucleotides (TCATTCAG) from intron 11 upstream of the wildtype splice acceptor of exon 12 (Figure 2D). Due to the low resolution of the gel electrophoresis, the size difference of the 8 nucleotides in the main PCR product is not visible in both the minigene and blood cDNA (Appendix A). Sanger sequencing results from the mother’s sample revealed the presence of both the correct and the misspliced transcripts, in accordance with her carrier status (Figure 2D). In addition, evidence of exon 12-skipping was observed in cDNA derived from both the index patient and his mother (Figure 2D and Appendix A). This resulted in three overlapping Sanger sequences in the mother’s cDNA analysis (Figure 2D is indicated by an arrow).

## 4. Discussion

We report a novel non-canonical splice site variant in *RPGR* intron 11 identified in a patient with characteristic clinical manifestations of XLRP. Segregation analysis confirmed the carrier status in the patient’s mother, who showed a tapetal-like reflex, as assessed by FAF. In silico splicing prediction tools strongly suggested this variant weakens the wildtype splice acceptor and, concomitantly, creates a novel splice acceptor site in intron 11, 8 nucleotides upstream of exon 12. A functional assessment using a minigene assay showed an out-of-frame insertion of the respective 8 nucleotides of intron 11, ultimately leading to a premature stop codon.

PAX blood analysis confirmed the aberrant splicing previously observed in the minigene assay with a transcript analysis from the blood samples. Exon 12-skipping was observed in the minigene assay as well as in the blood’s mRNA, indicating that the minigene assay published by de Heer et al. is able to mimic proper RPGR splicing in vitro [43]. The skipping of exon 12 in the wildtype RPGR has been previously reported in the retina (approximately 0.8% of all isoforms), in the brain, in another minigene splicing assay, and in healthy human retinas from donors [16,44].

Canonical splice site variants are located in the dinucleotides adjacent to exons as these bases are highly conserved within human splice donors and acceptors [45] and thus are included in the PVS1 (very strong evidence of pathogenicity) category of the ACMG’s classification. In contrast, novel NCSS variants can only be classified as a VUS by the ACMG unless a functional analysis is performed. It has been reported that the contribution of the NCSS is underestimated by approximately 35–40% [45]. Thus, conducting functional analyses to assess the potential effects of candidate NCSS variants on splicing is of paramount importance.

In addition, several recent studies indicate the impact of the NCSS in molecular diagnostics for patients affected by retinal-inherited diseases. Pathogenic NCSSs have been described in many genes associated with inherited retinal dystrophies, including *ABCA4*, *BEST1*, *POC1B*, *CACNA2D4*, *FSCN2*, *MAK*, *MERTK*, *PRCD*, *RIMS1*, *RP2*, *RPGR*, or *USH2A* [44,46,47,48,49,50]. Moreover, a large study focusing on *ABCA4* has shown that sequencing the entire locus (including the intronic regions) allowed the identification of a molecular diagnosis in 42.5% of the probands with suspected *ABCA4*-related retinopathy [51]. The authors found that NCSS variants represented 10% of all unique pathogenic alleles identified, which highlights the importance of this variant type in inherited retinal dystrophies.

The results of the well-established minigene assay or whole blood mRNA analysis of a candidate splice site variant fulfills the PS3 category criterion of the ACMG’s classification. The variant reported here can thereby be classified as likely pathogenic (PS3, PP1, PP3, and PP4 categories of the ACMG’s classification) due to the mRNA transcript analysis results obtained with both assays.

In addition, the ocular phenotype in the mother supports our classification of the intronic variant in the *RPGR* as potentially disease-causing. The clinical findings in the mother are characteristic for female carriers with pathogenic *RPGR* variants and, therefore, also argue against the potential pathogenicity of the variant in *NRL* found in the index (Table 1).

Even though the phenotype associated with the gene *CC2D2A* is different from the clinical findings in the index and the splicing prediction software results were unremarkable, caution has also been taken when excluding this variant. In fact, this variant lies one nucleotide in front of the splice donor site and might weaken the donor and activate a cryptic donor site. A similarly located variant in the gene *CHD7*, which is located two bases upstream of the splice donor site, was described by Haug et al. to alter splicing by activating an alternative donor site [29]. Moreover, an exon splice enhancer (ESE) or silencer (ESS) might also be affected by the variant and lead to different splicing patterns [52,53,54].

The feasibility of splicing correction by gene therapy has been shown for an *RPGR* donor splice site variant with the use of therapeutic U1snRNA [55]. The variant presented in this study, however, creates a novel splice acceptor site and, concomitantly, weakens the wildtype splice acceptor site. Thus, the use of a variant-adapted U1 snRNA would not suffice. Hence, gene replacement therapy would be favored in this case: nonsense-mediated RNA decay (NMD) or degradation of the predicted truncated protein is expected. This was previously shown for the nonsense variant NM_000328.3:c.1154T>A (p.Leu385*) [56]. The authors reported that the truncated RPGR protein could not be detected at the primary cilium in patient-derived fibroblasts.

This study emphasizes the contribution of non-canonical splice variants in molecular diagnostics and the requirement of functional assays for the re-classification of those types of variants. It seems that a minigene assay is a reliable approach to characterize NCSS variants in the *RPGR* if no patient blood sample for RNA extraction is available. In addition, the splice effects of a variant need to be carefully evaluated on the transcript (cDNA) level for designing future therapeutic approaches.

## Figures and Tables

**Figure 1 genes-14-00934-f001:**
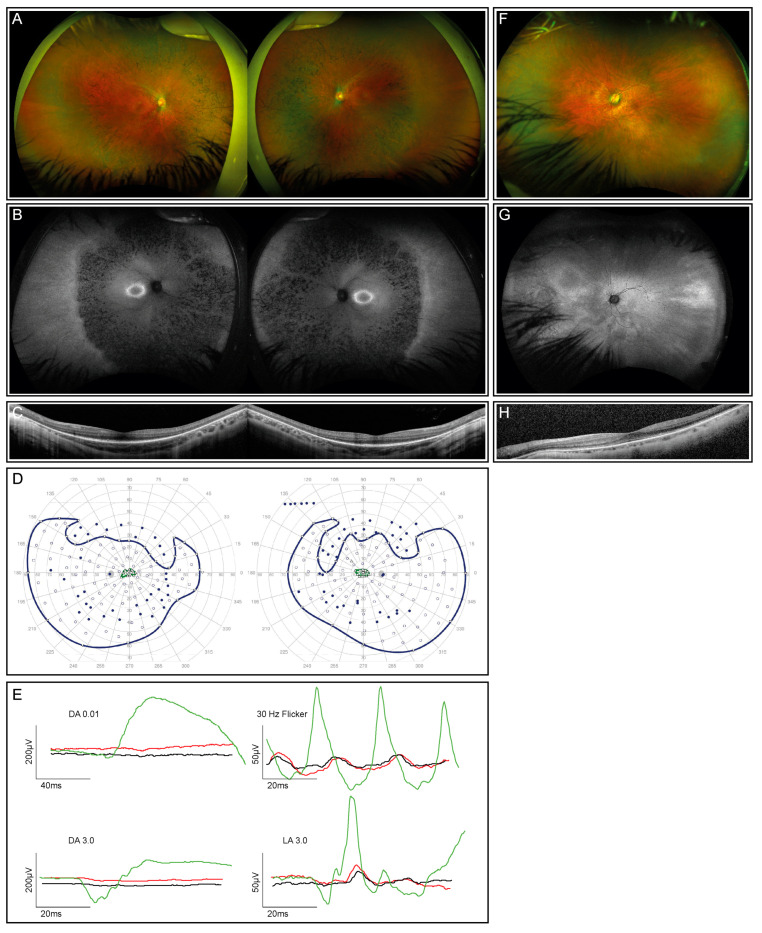
Clinical phenotype for the index patient (both eyes) and his mother (left eye only; right eye not shown due to dense cataract and consequently insufficient image quality). (**A**) Fundus images of the index patient demonstrate advanced retinal dystrophy with bone spicule pigmentation. (**B**) Corresponding autofluorescence images show retinal atrophy and a circular ring of increased autofluorescence. (**C**) OCT scans through the macula of the index patient reveal reduced outer segment layers. (**D**) Kinetic visual fields (only available from the index patient) showed a severely constricted field to isopter I:4e and relatively preserved temporal and lower fields to the large isopter V:4e (filled circles: test target not seen, unfilled circles: test target seen). (**E**) Full-field ERG responses of both eyes of the index patient (red, right eye; black, left eye) in comparison with an age-matched normal subject (green traces). DA, dark-adapted, LA, light-adapted. All responses of the patient are severely reduced and delayed. (**F**) The fundus image of the mother’s left eye showing reduced retinal sheen but no bone spiculae pigmentation. (**G**) Corresponding autofluorescence image of the mother’s left eye showing a tapetal-like reflex with a patchy and nasally speckled pattern. (**H**) OCT scan through the macula of the mother’s left eye with a reduced outer segment layer of lesser extent than her son’s.

**Figure 2 genes-14-00934-f002:**
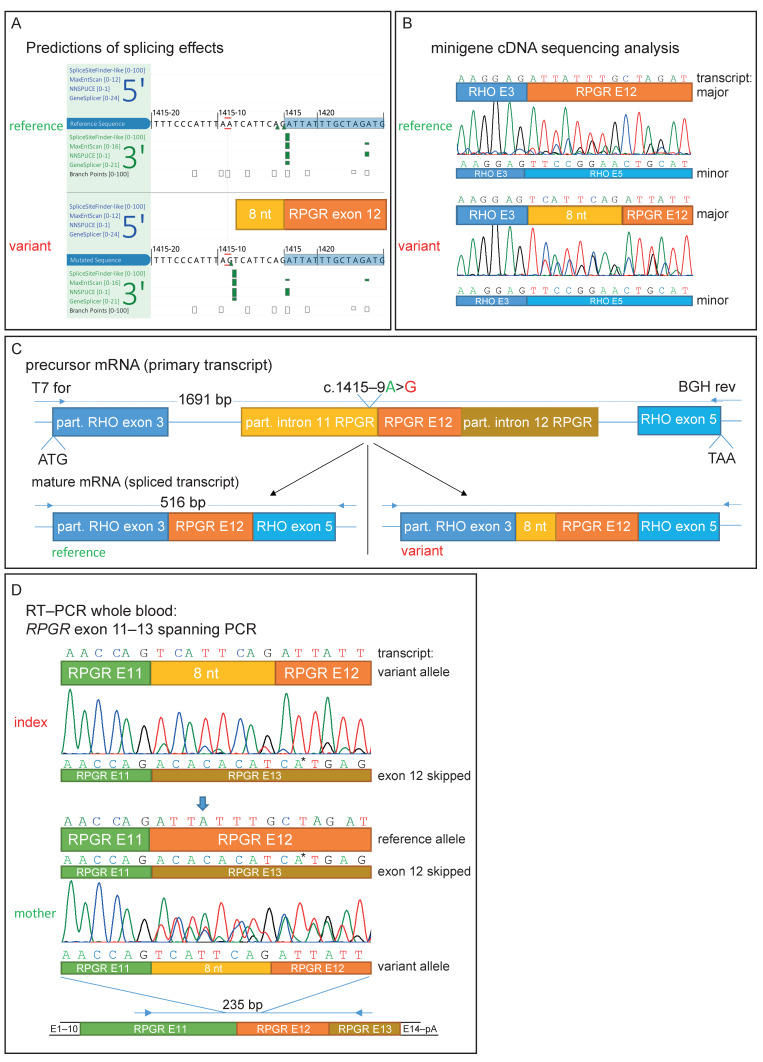
Genetic testing and functional characterization of the sequence variant. (**A**) Prediction of different splice algorithms, shown by AlamutVisualPlus, predicted a new acceptor site 9 nucleotides upstream of exon 12. (**B**) Sequence analysis of transcripts derived from the minigene assay. Analysis showed a predicted 8 nt extension of exon 12 by the found variant and, in addition, exon 12-skipping. (**C**) Minigene construct used for *RPGR* NM_001034853.1:c.1415-9A>G splice effect analysis. A part of *RHO* exon 3 to partial exon 5 was cloned into the pcDNA3.1 vector. *RHO* exon 4 was replaced by *RPGR* exon 12 with flanking introns. Predicted spliced products are shown for the reference and variant transcript. (**D**) RNA was extracted from whole blood samples from the patient and his mother. Similar observations as in the minigene sequencing analysis were observed. Arrow shows the only position in the depicted nucleotide stretch with three different bases (from three different transcripts: reference, exon 12-skipping, and exon 12 extended by 8 nucleotides). * Sequencing artifact: this marked base derived from exon 13 of the *RPGR* was only seen in the reverse sequencing chromatogram.

**Table 1 genes-14-00934-t001:** Variants obtained after filtering WES data from the index patient. The selection criteria, in addition to the frequency, are highlighted in bold letters in the table.

Gene	cNomen	gnomAD (%)	CADD PHRED	Dist. NSS	LSE	Splice AI (Score)	HGMD	ACMG	Zyg
*CC2D2A*	NM_001080522.2: c.1017C>T	**0.0008**	0.0	**−1**		DL (0.05)	-	LB	het
*GRM6*	NM_000843.4: c.138G>A	**0.0822**	12.1	154	**NAS**	AG (0.17)	-	LB	het
*TMEM67*	NM_153704.6: c.2848G>A	**0.0008**	**28.3**	−60		all (0.00)	DM?	VUS	het
*INPP5E*	NM_019892.6: c.1159+8C>T	**0.4092**	0.4	**8**		DL (0.02)	-	B	het
*NRL*	NM_006177.3: c.14C>G	**0.0004**	**23.7**	41		all (0.00)	-	VUS	het
*RPGR*	NM_001034853.2: c.1415-9A>G	**-**	**23.9**	**−9**	**NAS**	AG/AL (0.98)/(0.95)	-	VUS	hem

gnomAD (%): minor allele frequency percentage, as reported by gnomAD “all”; Dist. NSS: distance nearest splice site; LSE: Alamut local splice site effect (average change predicted by MaxEntScan, NNSPLICE, and SSF); NAS: new acceptor site; DL: donor loss; AG: acceptor gain; AL: acceptor loss; all: donor loss, donor gain, acceptor loss, and acceptor gain; HGMD: Human Gene Mutation Database classification; DM?: disease-causing mutation? (HGMD classification); ACMG: American College of Medical Genetics and Genomics classification, classified by varsome.com (accessed on 17 March 2023); LB: likely benign; VUS: variant of unknown significance; B: benign; Zyg: zygosity.

## Data Availability

The data presented in this study are available on request from the corresponding author.

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
