# Peer review of "Functional Analysis of a Novel, Non-Canonical RPGR Splice Variant Causing X-Linked Retinitis Pigmentosa"

_genes, 2023, doi:10.3390/genes14040934_

Round 1

Reviewer 1 Report

Koller et al. realized a very interesting article describing the "Functional analysis of a novel, non-canonical RPGR splice variant causing X-linked retinitis pigmentosa". I consider the manuscript very interesting but, at the same time, I suggest several revisions needed to improve the reliability and the completeness of the paper:

1) The "Introduction" section should be more updated and improved. I suggest adding data related to recent approaches to identify novel mutations and related alteration of molecular pathways in retinal dystrophies patients by using innovative NGS analyses pipelines. The recent PMID: 36290689 and PMID: 36490268 could represent a substrate able to enforce the role of considered cellular mechanisms.

2) The "Methods" section should be improved, especially in the part related to NGS analysis, which has to be more detailed. Then, why did the authors use the hg19 reference genome instead of the more recent hg38?

3) Finally, manuscript requires important English revisions and typos correction

Author Response

REVIEWER 1 comments:

 Koller et al. realized a very interesting article describing the "Functional analysis of a novel, non-canonical RPGR splice variant causing X-linked retinitis pigmentosa". I consider the manuscript very interesting but, at the same time, I suggest several revisions needed to improve the reliability and the completeness of the paper:

  • The "Introduction" section should be more updated and improved. I suggest adding data related to recent approaches to identify novel mutations and related alteration of molecular pathways in retinal dystrophies patients by using innovative NGS analyses pipelines. The recent PMID: 36290689 and PMID: 36490268 could represent a substrate able to enforce the role of considered cellular mechanisms

We thank you for your valuable time and the valuable comments and inputs.

The introduction has been revised to include sections discussing the clinical and genetic characteristics of the disease retinitis pigmentosa as well as additional details on the RPGR gene and protein. Also, details on pathogenic variants in the RPGR gene have been added.

  • The "Methods" section should be improved, especially in the part related to NGS analysis, which has to be more detailed. Then, why did the authors use the hg19 reference genome instead of the more recent hg38?

The NGS analysis pipeline has been described in more detail in the revised manuscript, including packages versions, tools applied in the workflow and variants filtering strategy. Due to the more detailed workflow, we have also updated the results section by including a table showing all the variants which were identified with our filtering strategy.

While we agree that it may be an alternative to update the workflow to the human reference genome hg38, our laboratory is accredited for molecular diagnostics with specific pipelines. Changing to hg38 would require new validation procedures. Moreover, it has been shown that there are important discrepancies in variant calling sets from the same raw WES data when using either hg19 or hg38 references. However, no difference was identified in any gene associated with inherited retinal dystrophies (Li et al. 2021, https://doi.org/10.1016/j.ajhg.2021.05.011).

  • Finally, manuscript requires important English revisions and typos correction

We have worked at improving the language and a few typos. If specific phrases or paragraphs of the manuscript are unclear, we would appreciate a more detailed input (i.e. line number) so that we can more readily address the problem.

Reviewer 2 Report

In this study, Koller et al. reported a novel splicing mutation c.1415- 9A>G in a RP patient and his mother. The mutation leads to a frameshift and a premature stop codon in RPGR transcripts. This mutation has not been reported and be likely pathogenic​.

The following suggestions might be of use to the authors.

1. The sequence chromatogram in fig2B and 2D is difficult to read. Using T-Vector to get the monoclone and then sequencing can resolve this issue.

2. In fig2C, the upper diagram, why the intron 11 is behind exon12? Please also include the diagram of exon12 skipping.

3. Please indicated the location of primers in the diagram in Fig2C.

4. It is seemly no difference in the mw of bands in figS1B and S1C. High resolution agarose gel electrophoresis or PAGE may be an option.

Author Response

REVIEWER 2 comments:

In this study, Koller et al. reported a novel splicing mutation c.1415- 9A>G in a RP patient and his mother. The mutation leads to a frameshift and a premature stop codon in RPGR transcripts. This mutation has not been reported and be likely pathogenic.

 The following suggestions might be of use to the authors.

  • The sequence chromatogram in fig2B and 2D is difficult to read. Using T-Vector to get the monoclone and then sequencing can resolve this issue.

Thank you for your valuable time and carefully reviewing our manuscript.

We are aware that overlapping frameshift Sanger sequencing data are difficult to read. We have addressed this issue in the manuscript by written letters in the corresponding color of the chromatogram. In addition, by showing overlapping Sanger sequences, the lower chromatogram peak heights and areas show the existence of minor splice products. We have pointed this out in the figure accordingly by adding “minor” and “major transcript”.

Splicing is a dynamic process and thus splice products are not homogenous. Sub-cloning cDNA of splice products as suggested to retrieve single sequences would lead to the risk of losing minor transcripts or a large number of clones would need to be screened in order to obtain clones containing minor transcripts.

In addition, the advantage of showing non-monoclonal Sanger sequencing data of cDNA is that an approximate ratio of the different transcript (areas of the individual peaks) can be visualized.

  • In fig 2C, the upper diagram, why the intron 11 is behind exon12? Please also include the diagram of exon12 skipping.

 Thank you for this important point. We have added the correct labelling in the updated figure. Exon 12 skipping is now more clearly indicated in the updated figures 2B and C.

  • Please indicated the location of primers in the diagram in Fig2C.

Thank you for this suggestion. We have added an in-scale scheme in figure 2 C showing the primer binding sites. 

  • It is seemly no difference in the mw of bands in figS1B and S1C. High resolution agarose gel electrophoresis or PAGE may be an option.

The intention of showing the PCR fragments in figures S1B and C is not to show the 8 nt difference in size between the variant and the reference transcripts but rather the approximate amounts.

In figure S1B we wanted to show that exon 12 is skipped in some transcripts. On the other hand, figure S1C aims at highlighting that RPGR is also expressed in whole blood. The 8 nt extension of the PCR fragment is shown by Sanger sequencing in Figures 2 B and D.

To address your point and clarify our intention with Fig. S1B and C we added the following paragraph: “Interestingly, Sanger sequencing revealed partial RPGR exon 12 skipping as can be seen by overlapping sequences in both, the variant and reference constructs. This was also detected by gel electrophoresis as shown by a smaller fragment of 92 bp in size and also in lower amounts (Fig. 2B and S1B). It is important to note that the correctly spliced transcript (without additional 8 nt) could not be detected by Sanger sequencing in the spliced transcripts of the variant minigene construct.” and “We were able to amplify RPGR exons 11 to 13 in cDNA derived from a peripheral blood sample from the index patient and his mother (Fig. S1C), confirming that RPGR is expressed at low levels in whole blood, as reported in the GTEx database.”

Round 2

Reviewer 2 Report

I have no further comments.